# Risk factors for neonatal hypothermia at Arba Minch General Hospital, Ethiopia

**Tegenu Tessema[1], Tilahun Ferede Asena[ID][2]\*, Meseret Mosissa Alemayehu[3], Asmare Mekonnen Wube[4]**

**1** Department of Statistics, College of Natural Sciences, Jinka University, Jinka, Ethiopia, **2** Department of Statistics, College of Natural Sciences, Arba Minch University, Arba Minch, Ethiopia, **3** Department of Pediatrics, Pawe General Hospital, Pawe, Ethiopia, **4** Ethiopian Public Health Institute, Addis Ababa, Ethiopia

\* feredetilahun14@gmail.com

## Abstract

### Background

The first few minutes after birth are the most dangerous for the survival of an infant. Babies in neonatal intensive care units are either under heated or overheated, and hypothermic infants remain hypothermic or develop a fever. As a result, special attention must be paid to monitoring and maintaining the time of recovery from hypothermia states. Despite numerous studies, only a few have examined the transition from neonatal hypothermia and associated risk factors in depth.

### Method

A retrospective observational study was conducted to track axillary temperatures taken at the time of neonatal intensive care unit admission, which were then tracked every 30 minutes until the newborn's temperature stabilized. All hypothermic neonates admitted to the neonatal intensive care unit between January 2018 and December 2020 was included in the study. Temperature data were available at birth and within the first three hours of admission for 391 eligible hypothermic neonates. The effect of factors on the transition rate in different states of hypothermia was estimated using a multi-state Markov model.

### Result

The likelihood of progressing from mild to severe hypothermia was 5%, while the likelihood of progressing to normal was 34%. The average time spent in a severe hypothermia state was 48, 35, and 24 minutes for three different levels of birth weight, and 53, 41, and 31 minutes for low, moderate, and normal Apgar scores, respectively. Furthermore, the mean sojourn time in a severe hypothermia state was 48, 39, and 31 minutes for three different levels of high, normal, and low pulse rate, respectively.

### Conclusion

For hypothermic survivors within the first three hours of life, very low birth weight, low Apgar, and high pulse rate had the strongest association with hypothermia and took the longest

**Data Availability Statement:** All relevant data are within the manuscript and its Supporting Information files.

**Funding:** The authors received no specific funding for this work.

**Competing interests:** The authors have declared that no competing interests exist.

**Abbreviations:** AIC, Akaike's Information Criterion; APGAR, Appearance Pulse Grimace Activity Respiration; CSA, Central Statistics Agency of Ethiopia; LBW, Low Birth Weight; MSM, Multi State Model; NICU, Neonatal Intensive Care Unit; VLBW, Very low Birth Weight; WHO, World Health Organization.

time to improve/recover. As a result, there is an urgent need to train all levels of staff dealing with maintaining the time of recovery from neonatal hypothermia.

## 1. Background

Neonatal hypothermia is defined as a core body temperature of less than 36.5˚C. Because it is the transitional period from intrauterine to extra uterine life, the first few minutes after birth are the most dangerous for infant survival [1]. Sustained body temperature decrease increases the metabolic demands of the neonate and has been linked to sepsis, asphyxia, respiratory distress syndrome, and mortality [2]. As a result, caregivers must work hard to prevent neonatal hypothermia in the first few minutes after birth.

According to WHO, when a baby's heat loss exceeds his or her ability to produce heat, the baby's body temperature falls below the normal range (36.5˚C—37.5˚C). The newborns are suffering from mild hypothermia, with temperatures ranging from 36.0˚C to 36.4˚C, which should be cause for concern. A body temperature of 32.0˚C to 35.9˚C is considered moderate hypothermia. A newborn with a temperature of less than 32.0˚C is considered to be suffering from severe hypothermia and should be treated as soon as possible [1]. Infant rewarming after birth should be optimized and time allotted for it to reduce the presence of hypothermia after birth even before undergoing procedures at the neonatal intensive care unit [3].

Thermal care is essential for reducing newborn morbidity and mortality. The monitoring system, however, is insufficient, and sensors occasionally detach or are not observed by health professionals, babies in neonatal intensive care units are under heated or overheated, and infants with hypothermia at birth and admission remain hypothermic or develop fever [4]. Neonatal hypothermia, regardless of climate, is a major cause of neonatal death and health impairment [5]. Because of their high surface area per unit of body weight, newborns are unable to fully maintain their body temperature. A neonate's skin temperature can drop at a rate of 0.1˚C to 0.3˚C per minute if no action is taken immediately after birth [6]. As a result, special precautions must be taken in neonatal intensive care units to monitor and maintain newborns' time of recovery from hypothermia.

The prevalence of hypothermia at the hospital ranged from 32% to 85%, with rates varying even in tropical environments [7]. Approximately 81,000 babies die in Ethiopia during their first four weeks of life. This accounts for 42% of all deaths among children under the age of five. Ethiopia had a prevalence of postnatal hypothermia of 69.8% [8]. Infants with birth weights ranging from 1000 to 1499 grams and mild hypothermia (36.0˚C-36.5˚C) died at a rate of 40.8%, while those with temperatures below 34.0˚C died at a rate of 56.8% [2]. Risk factors are inextricably linked to newborn health, and illness has a significant impact on neonate health and survival [9].

Several studies have been conducted to determine the prevalence and causes of hypothermia. However, few have thoroughly addressed the transition from a hypothermic state and the risk factors that accompany it. Iranian researchers discovered a link between some hypothermia risk factors, such as birth weight and Apgar score, but no link between environmental temperature and hypothermia [10]. The other study looked into whether keeping newborns in a suitable thermal environment accelerated the transition from hypothermia to normalcy [11].

Despite this, the study took temperature measurements in the first two hours, did not take into account neonatal observation transitions to the severe hypothermia state in the first three hours, and included some other risk factors such as pulse rate, respiratory rate, and oxygen

saturation. Concerns about the scarcity of long-term studies on the transition from hypothermia prompted the researchers to look into the subject. A patient's experience in a survival study can be thought of as a two-state process with one possible transition from a 'live' to a 'dead' state [12]. In this study, however, the "hypothermia" state was divided into two transient states, each of which corresponded to a different stage of the illness. When a newborn is in one of a set of discrete states at any given time, multi-state models can be used to model state transitions. It should be noted that the majority of research on neonatal hypothermia transition has concentrated on risk factors. The ability to quantify disease dynamics, such as the time it takes to reach a specific state or the likelihood of movement, may lead to more effective disease prevention, management, and treatment. We wanted to look at a shorter time span with observed covariates. As a result, the goal of this study was to bridge that gap by analyzing data from hypothermic neonates at Arba Minch General Hospital, as well as to identify other risk factors associated with the rate of transition between different states of neonatal hypothermia.

## 2. Methods

### 2.1. The data

Retrospective data were obtained from Arba Minch General Hospital for the period from January 2018 to December 2020. The longitudinal measurement data on the record of neonate patient care follow-up cards were used in this study at regular time intervals. The standardized clinical recommendations for maintaining an infant's normal body temperature, which is one of the fundamental components of neonatal care, are to take temperature measurements every 30 minutes [1].

### 2.2. Study design

This retrospective study examined newborns with up to seven surveillance scans in critically ill babies at the neonatal intensive care unit. Before data collection began, the institutional review board approved the study protocol. Data was extracted from the cards of hypothermic neonates by identifying the neonates' cards using their respective Medical Record Numbers from the medical center, and the data extraction was completed in the card room per the researcher's agreement. Following data extraction, data entry, editing, coding, and organization were completed. R software version 4.1.1 was used to conduct descriptive statistics. MSM (Version 1.6.8) was used in R to perform statistical inference.

### 2.3. Sampling design

The study protocol was approved by the institutional review board before data collection began. Data was extracted from hypothermic neonates' cards by identifying the neonates' cards using their respective Medical Record Numbers from the medical center, and the extraction was completed in the card room per the researcher's agreement. Data extraction was followed by data entry, editing, coding, and organization. To conduct descriptive statistics, R software version 4.1.1 was used. Fig 1 shows the conceptual framework of sampling procedure.

### 2.4. Inclusion and exclusion criteria

The newborn babies diagnosed with hypothermia at birth, observed at a minimum of three different time points, had an Apgar score recorded, and inborn admitted to NICU on the same day were included but outborn babies, newborns babies who remain hypothermic for more than three hours and who died without gaining temperature within three hours were excluded. An exit criterion was the infant getting to a normal state.

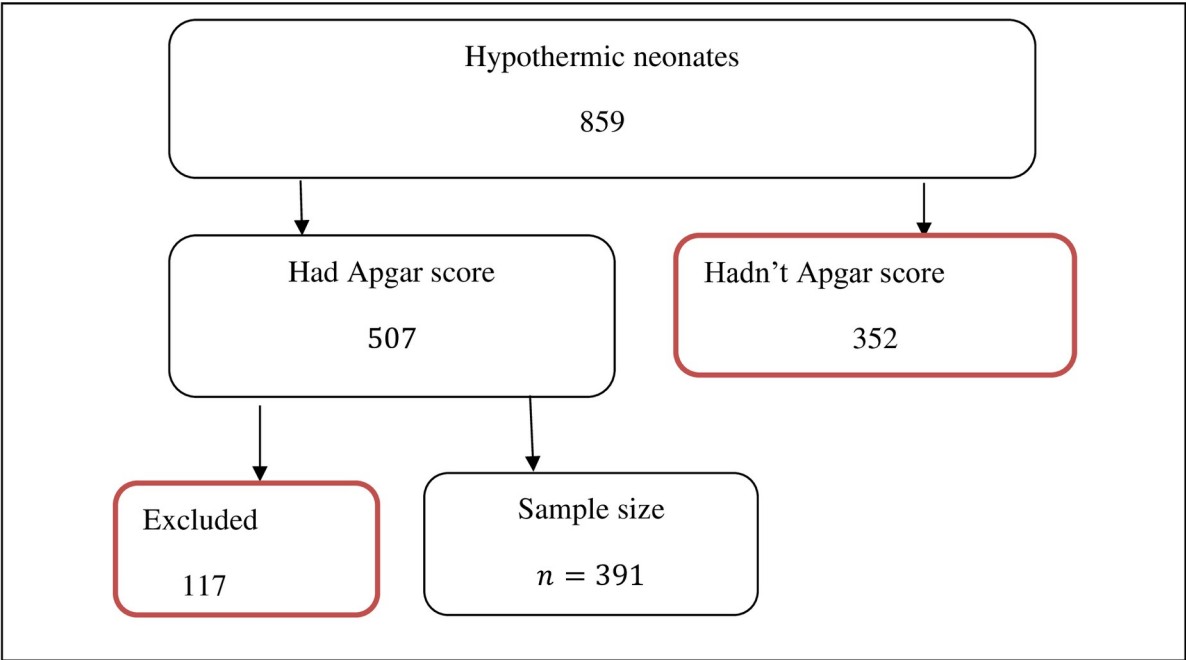

**Fig 1. The conceptual framework of sampling procedure.**

## 2.5. Variables in the study

Hypothermia severity was classified as severe (32.0˚C), moderate (32.0–35.9˚C), mild (36.0–36.4˚C), and normal (36.5–37.5˚C) [1]. In this study, the first two groups were combined and classified as severe hypothermia (state I), mild hypothermia (state II), and normal body temperature (state III) [11]. Gender (male, female), birth weight (very low birth weight (1500gram), low birth weight (1500-2500gram), normal birth weight (>2500gram) [1]), Apgar score at 5 minute (low (0–3), moderate (4–6), normal (>2500gram) [1]) were independent variables (7–10) [13]) Pulse rate (lower rate (80 bpm), normal rate (80–160 bpm), higher rate (>160 beats per minute) [14]), Respiratory rate (lower rate (30 bpm), normal (30–60 bpm), faster (> 60 birth per minute) [1]) and newborn oxygen saturation (low (90%), normal (90–95%), high (> 95%) saturation [1]). The explanatory variables were chosen from the literature and were all included in the models.

## 2.6. Operational definitions

**Hypothermia**: an axillary temperature of a newborn baby less than 36.5˚C

 **The respiratory rate:** is the rate at which breathing occurs

 **Pulse/heart rate:** is the wave of blood in the artery created by contraction of the left ventricle during a cardiac cycle

 **Oxygen saturation** is the fraction of oxygen-saturated hemoglobin relative to total hemoglobin in the blood.

## 2.7. Data analysis method

We employed a discrete time method, to represent movement between hypothermia states, a multi-state Markov model with constant transition rates is used. The normal state was thrilling because it does not allow for exits. We focused on the first occurrence of the normal state

("first hitting time") due to the short duration of the study and ignored further potential recovery. Severe and mild hypothermia were both transient conditions.

Longitudinal data consisted of observations of the disease process at arbitrary times. The exact times at which state transitions occurred were not known although the underlying process evolved continuously in time. Kalbfleisch & Lawless (1985) introduced the analysis of panel data under a Markov assumption where movements between disease states are governed by transitional intensities $q_{ij}(t, z(t))$ with i, j = 1, 2, 3 (the three possible states) and depend on time t and individual level or time-dependent explanatory variables at time t, denoted z(t) [15]. In our case, the $q_{ij}$ forms a (3x3) matrix Q whose rows sum to zero so that the diagonal entries are defined by $q_{ij} = -\Sigma_{i \neq j} q_{ij}$ Since it is not possible to move from state III to either state I or state II, then in the transition intensity matrix, $q_{31} = q_{32} = q_{33} = 0$.

Marshall and Jones (1995) introduced a new class of models by restricting the number of parameters, allowing all progressive or all regressive transitions to have the same regression coefficients, that is, $q_{ijl}(t) = \begin{cases} q_{ij}(0) \, e^{\beta_p z_l} & j = i+1 \\ q_{ij}(0) \, e^{\beta_r z_l} & j = i-1 \end{cases}$ An even more restrictive model can be introduced by setting the regression coefficients to $\boldsymbol{\beta}_p = -\boldsymbol{\beta}_r = \boldsymbol{\beta}$ [16]. Using the technique of regression, many covariates can be incorporated such as time varying covariates [17]. Details of the model description is found at supplementary file

In multi-state models, data are considered as series of observations $x_{i0}, x_{i1}, \ldots, x_{in}$ and at times $t_{i0}, t_{i1}, \ldots, t_{in}$ which is the product of $X(t)$ process. In this process the amount of $1, \ldots, I$ states is $i = 1, \ldots, N$ for each newborns, with covariate vectors $z_l$ and model parameters $\theta$, the log-likelihood under a Markov assumption can be expressed as [18] $L(\boldsymbol{\theta}) = \Sigma_{i=1}^{N} \Sigma_{j=1}^{ni} \log(p_{x_{i(j-1)} x_{ij}}(t_{i(j-1)}, t_{ij}; z_l, \boldsymbol{\theta}))$ where, $p_{ij}(t_0, t_1, \boldsymbol{\theta}) = p(x(t_1) = j | x(t_0) = i; \boldsymbol{\theta})$ and $(i, j)$ Entry of $I \times I$ matrix is the transition probability which can be found by solving Kolmogrov Forward equation [19]. Details of the model description is found at supplementary file

## 2.8. Model comparisons

For these models, the Akaike's Information Criterion (AIC) Akaike (1987) was used for model selection: $AIC = -2ln(likehood) + 2p$ where p is the number of parameters in the model and n is the number of subjects in the data or sample size. Detailed description is found as supplementary file

## 2.9. Model diagnostics

The likelihood ratio test statistic was also used to assess the models' time inhomogeneity [20] Aguirre-Hernandez and Farewell (2002) developed the Pearson-type goodness-of-fit test for the hypothesis that longitudinal data was generated by a fitted Markov model [21].

## 2.10. Ethical approval and consent

The Arba Minch General Hospital's research advisory board has given its written approval and consent (Ref.No.AMGH/11309/13) for the use of the neonates' data in the study (Ref. No. AMGH/11309/13). There were no links to neonates, and no personal information were included in the data directly for privacy reasons. Arba Minch University research and ethics committee also approved the research protocol (Ref. No. stat/519/2012). The methods used in participant-involved studies were carried out in accordance with the ethical standards established by national and institutional research committees.

## 3. Results

The study included 391 neonates with hypothermia in total. There were 187 female infants and 204 male newborns in this sample (52.2%). There were 120 newborns with normal birth weight, 246 (60.3 percent) with low birth weight, and 35 with extremely low birth weight, according to the diagnoses of newborns admitted to the hospital. For neonates admitted to the NICU, the average, typical, and median length of stay were, respectively, 6, 4, and 5. About 27 (6.9 percent) of the newborns admitted to the NICUs died before being released from the hospital, while 45 were sent to other hospitals and discharged against medical recommendation. 3919 of the babies brought to the NICUs recovered and were sent home, while 27 (6.9 percent) passed away before leaving the hospital and 45 were sent to other hospitals and released against medical recommendation (self-discharges). Table 1 shows the stages of hypothermia in infants at seven different observational time points. As demonstrated in Table 1, 176 neonates suffer severe or moderate hypothermia at birth, while 215 babies (55%) have mild hypothermia.

### 3.1. State transition between different possible states

Table 2 shows about 1883 longitudinal observations from the 391 babies admitted to the NICU who recovered from hypothermia in the first three hours as part of the trial. On 85 times, severe hypothermia led to normal states, while mild hypothermia in 306 cases led to normal states. On 137 occasions, observations of severe hypothermia were recorded, then those of mild hypothermia. There were 46 instances of moderate hypothermia seen followed by severe hypothermia. The exclusion of several transitions based on how long it took to return to the normal state, such as the transition from the normal state to mild hypothermia for fewer than three hours after recovery, was supported by this finding.

### 3.2. The estimated transition intensities on the covariates

The transition intensities between neonatal hypothermia states from severe to moderate hypothermia and from mild hypothermia to normal state were calculated to be 0.027 and 0.020 per minute, respectively, using a fitted multi-state Markov model with three covariates. According to this, the change from severe to moderate hypothermia happened faster than the change from mild to normal hypothermia.

The transition intensities between neonatal hypothermia states from severe to moderate hypothermia and from mild hypothermia to normal state were calculated to be 0.027 and 0.020 per minute, respectively, using a fitted multi-state Markov model with three covariates. According to this, the change from severe to moderate hypothermia happened faster than the change from mild to normal hypothermia.

**Table 1. The states of hypothermia among newborns at Arba Minch General Hospital, Ethiopia, 2018–2020.**

| Observation time in minute | States of hypothermia | | |
|---|---|---|---|
| | Severe state | Mild state | Normal state |
| 1 | 176 | 215 | 0 |
| 30 | 105 | 178 | 108 |
| 60 | 60 | 136 | 87 |
| 90 | 40 | 81 | 75 |
| 120 | 26 | 59 | 36 |
| 150 | 15 | 10 | 60 |
| 180 | 0 | 0 | 25 |

**Table 2. Shows the transitions probability matrix computed from data with various states of neonatal hypothermia in the first three hours.**

| From / To | Severe | Mild | Normal |
|---|---|---|---|
| Severe | 376 (63%) | 137 (23%) | 85 (14%) |
| Mild | 46 (5%) | 542 (61%) | 306 (34%) |
| Normal | 0 (0%) | 0 (0%) | 391 (00%) |

The hazed ratio was 0.98, or 2% lower risk for the normal condition compared to mild hypothermia, with a 95 percent confidence interval spanning from 0.82 to 1.17. The hazed ratio was 1.50, or 50% higher risk for mild hypothermia compared to severe hypothermia. Accordingly, neonates with normal birth weights have a higher chance of changing their condition than newborns with extremely low birth weights. With a 95% confidence interval of 1.06 to 1.59 and 1.14 (14% higher risk for moderate hypothermia against severe hypothermia), respectively, the hazed ratio for different levels of Apgar score was 1.30 or 30% higher risk for normal state than mild hypothermia. Accordingly, neonates with normal Apgar scores had a higher likelihood of changing states than newborns with low Apgar scores. At various pulse rate levels, the hazed ratio was calculated to be 1.03 or 3% higher risk for normal state compared to mild hypothermia and 0.48 (51%) lower risks for severe hypothermia states compared to mild hypothermia states, with a 95 percent confidence range spanning from 0.24 to 0.97. According to this, infants with greater heart rates were less likely to experience state changes than infants with lower heart rates.

## 3.3. Estimated average time spent in transient state

Table 3 shows that for neonates with very low birth weights, the average duration spent in severe and mild hypothermia was roughly 48 and 40 minutes, respectively, depending on the effect of weight on the transition rate from severe to mild hypothermia. Infants with low Apgar scores spent about 53 and 46 minutes, respectively, sojourning in severe and mild hypothermia, according to the effect of Apgar scores on transition rates from mild hypothermia to normal state.

The average time spent in severe and mild hypothermia for neonates with higher heart rates who were at higher risk was 48 and 34 minutes, respectively, according to the effect of pulse rate on transition rate from mild to severe hypothermia. This shown that the average

**Table 3. Shows the estimated average times spent in severe and mild hypothermia for each group of newborns based on birth weight, Apgar score, and pulse rate.**

| Covariate name | Covariate levels | Estimated average times spent in severe hypothermia (minute) | Estimated average times spent in mild hypothermia (minute) |
|---|---|---|---|
| Birth Weight | VLBW | 48 | 40 |
| | LBW | 35 | 38 |
| | Normal | 24 | 36 |
| Apgar score at 5 minutes | Low | 53 | 46 |
| | Medium | 41 | 42 |
| | Normal | 31 | 36 |
| Pulse rate of newborns | Lower | 31 | 34 |
| | Normal | 39 | 40 |
| | Higher | 48 | 42 |

time in transient states may be longer for one group than the other, depending on the risk factors and the hypothermic neonates.

### 3.4. Results for model comparison

The covariate model's maximum likelihood of the unknown parameters had a lower AIC score of 2118.98, which was better. The measurements' results demonstrated that, given that it best matches the available data, the selected covariates model can forecast future observations.

### 3.5. Model diagnostic check

Estimated expected probability of remaining in a specific state versus average time spent for severe hypothermia (top) and mild hypothermia (bottom) (bottom). Mild hypothermic newborns have spent more time in hypothermia than severe hypothermic newborns in order to recover. Fig 2 shows that the probability of entering a better state after 60 minutes with severe hypothermia was approximately 0.6 and 0.4 with mild hypothermia. This finding implies that the expected probability of remaining in mild hypothermia was higher when slow rewarming was used rather than rapid rewarming when treating severe hypothermic newborns.

The null hypothesis of the models' temporal inhomogeneity was disproved by the likelihood ratio test (p-value = 0.00016), and the covariate model offered a considerable improvement over the null model. The Pearson-type test did not reject the null hypothesis that longitudinal data fit the Markov property, and the chi-square statistics (34.71) and p values (0.53) show that these models offer a good overall fit. It would be predicted that a design that enabled factors to influence transition intensities would offer a better fit.

## 4. Discussions

Hypothermia can be induced in minutes, and the average time spent in hypothermic states is determined by risk factors and hypothermic neonates. The neonatal risk factors of very low birth weight, low Apgar score, and high pulse rate increased the mean sojourn time of

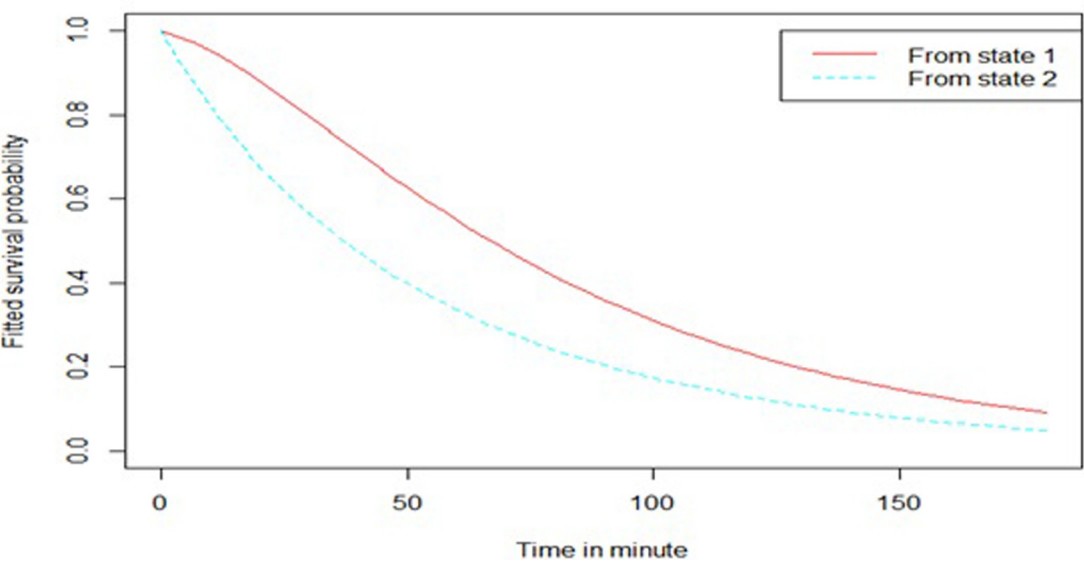

**Fig 2. Time-plot of the anticipated likelihood of staying in a particular state.**

newborns remaining hypothermic during the first three hours of postnatal life. This study backs up previous findings that neonatal hypothermia increases morbidity and hospital stay [22]. Neonatal hypothermia is linked to a higher risk of long-term morbidity [23]. This demonstrates that care providers should continue to consider temperature regulation during the first three hours of postnatal life to be a neonatal risk factor worthy of the attention of all newborn care providers. In this study, we focused on the first occurrence of normal state ("first hitting time") within three hours and ignored any subsequent potential recovery/loss. The probability of progressing from mild to severe hypothermia was 0.05. There were no transitions to a severe hypothermia state [10, 11, 24]. The difference could be explained by the small sample size of newborn babies with severe hypothermia.

We confirmed previous research findings that hypothermia causes clinical irreversible damage ranging from metabolic problems to neonatal death; therefore, a rapid transition from hypothermia is critical. The transition from severe to mild hypothermia occurs more quickly than the transition from mild to normal hypothermia [11, 24]. This could be because the World Health Organization recommends that healthcare providers consider rapid rewarming for severely hypothermic newborns and slow rewarming for mildly hypothermic newborns returning to normal [1]. We also found a link between risk factors that accelerate this transition and a positive step toward preventing post-hypothermic problems.

The average length of time spent separately in mild hypothermia for newborns with very low, low, and normal birth weight was 40,38,36 minutes in this study versus 38,29,22 minutes in Iran [11, 24]. The difference could be explained by weight having a significant effect on the rate of transition from mild hypothermia to a normal state, as well as the first hitting time of a normal state occurring within two hours.

The Apgar score influenced the transition from mild hypothermia to normal states in this study. Low Apgar newborns were more likely to die from hypothermia than normal Apgar newborns. In contrast, the Apgar score was not a risk factor for neonatal hypothermia transition [11, 24]. In the majority of studies [10, 23, 25–28], the Apgar score has been identified as a risk factor for neonatal hypothermia. Low Apgar score newborns spend more time in hypothermic states due to increased resuscitation efforts, prolonged management time in the delivery room, or increased inherent illnesses in these newborns. This could imply that preventing neonatal hypothermia during the first three hours after birth should be prioritized.

In this study, pulse rate had a significant effect on the rate of transition from mild to severe hypothermia. Hypothermia occurred at a lower rate in high pulse rate newborns than in low pulse rate newborns. A few studies have found that the newborn pulse rate is a risk factor for neonatal hypothermia. The most common symptom of hypothermia, on the other hand, is a decrease in the infant's pulse rate [29]. The heart cannot function normally when the body temperature drops. Assessing newborn infant pulse rates after birth is critical for directing recovery efforts in the right direction.

In this study, we include several risk factors for transitioning from hypothermic states and focus on estimating several quantities that can assist caregivers in reducing the risk of transitioning from neonatal hypothermia. The temperature of the NICU environment where the hypothermic newborns were born was not recorded. An earlier study suggested looking into the effect of birth temperature on the transition from hypothermia states. Their findings revealed that neonates born at temperatures higher than 28˚C spend less time in hypothermia [11, 24]. Furthermore, due to the abrupt change in ambient temperature, newborns are vulnerable to hypothermia [1].

A multi-state Markov model was used to investigate the transition rate and risk factors for transitioning from neonatal hypothermia states in several studies [11, 24]. However, the majority of studies have been cross-sectional or prevalence studies. This model may provide

researchers with a better understanding of the illness's process, allowing them to better understand how the disease evolves [9].

The retrospective observational design is one of the study's major limitations. Our findings should be interpreted in terms of a combination of severe and moderate hypothermia due to the small number of severe hypothermic newborns in our study. As a result, large numbers of severely hypothermic babies must be included in the study and analyzed separately. The use of a large number of newborns may allow for the detection of significant differences between hypothermia states to be more accurate. More research is required to determine the effect of risk factors on the progression of severe hypothermia to moderate hypothermia. Other study limitations included the exclusion of newborns who were persistently hypothermic or died, as well as the time interval between admission and recovery at the NICU within the first 24 hours.

## 4.1. Conclusion and recommendations

The exclusion of persistently hypothermic and deceased newborns limits these findings to less sick newborns. For hypothermic survivors within the first three hours of life, very low birth weight, low Apgar, and high pulse rate had the strongest association with hypothermia and took the longest to improve/recover.

These findings suggested that there is an urgent need to train all levels of staff dealing with monitoring and maintaining the time of recovery from neonatal hypothermia, particularly neonates with very low birth weight, low Apgar scores, and higher heartbeat. Future research should concentrate on deceased and persistent hypothermia, as well as the sickest newborns who were excluded.

## Supporting information

**S1 File.**
(RAR)

**S1 Data.**
(RAR)

## Acknowledgments

This study would not have been possible without the permission of Arba Minch University's Statistics Department. The official letter of cooperation referred as stat/519/2012 with subject, To Whom It May Concern was issued. The authors gratefully acknowledge the Neonatal Intensive Care Unit Department at Arba Minch General Hospital for allowing us to use their data in this study. We are grateful to the nurses who volunteered their time to collect data for our study.

## Author Contributions

**Conceptualization:** Tegenu Tessema, Tilahun Ferede Asena, Meseret Mosissa Alemayehu, Asmare Mekonnen Wube.

**Data curation:** Tegenu Tessema, Meseret Mosissa Alemayehu, Asmare Mekonnen Wube.

**Formal analysis:** Tegenu Tessema, Tilahun Ferede Asena.

**Investigation:** Tegenu Tessema, Meseret Mosissa Alemayehu, Asmare Mekonnen Wube.

**Methodology:** Tilahun Ferede Asena.

**Supervision:** Tilahun Ferede Asena, Meseret Mosissa Alemayehu, Asmare Mekonnen Wube.

**Validation:** Asmare Mekonnen Wube.

**Visualization:** Tegenu Tessema.

**Writing – original draft:** Tegenu Tessema.

**Writing – review & editing:** Tegenu Tessema, Tilahun Ferede Asena, Meseret Mosissa Alemayehu, Asmare Mekonnen Wube.

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
