## [Decision Letter · Decision Letter 0]

27 Sep 2022

PONE-D-22-10601Neonatal Hypothermia and Its Associated Risk Factors among Newborns at Arba Minch General Hospital, EthiopiaPLOS ONE

Dear Dr. Asena,

Thank you for submitting your manuscript to PLOS ONE. After careful consideration, we feel that it has merit but does not fully meet PLOS ONE’s publication criteria as it currently stands. Therefore, we invite you to submit a revised version of the manuscript that addresses the points raised during the review process.

The manuscript has been evaluated by three reviewers, and their comments are available below.

The reviewers have raised a number of concerns that need attention. They request additional information on the gap in the literature that this study addresses, and on methodological aspects of the study (such as quality control measures and how outcome variables are measured). The reviewers also request revisions to the discussion, to take into account the limitations of the study, and to the conclusions in order to take into account the implications of the results. 

Could you please revise the manuscript to carefully address the concerns raised?

We look forward to receiving your revised manuscript.

Kind regards,

Alice Coles-Aldridge

Editorial Office

PLOS ONE

Journal Requirements:

No

We would like to express our gratitude to Jinka and Arba Minch Universities for their financial support of this research.

No

6. We note you have included a table to which you do not refer in the text of your manuscript. Please ensure that you refer to Table 1,2,3,5 and 6 in your text; if accepted, production will need this reference to link the reader to the Table.

Reviewers' comments:

Reviewer's Responses to Questions

**Comments to the Author**

1. Is the manuscript technically sound, and do the data support the conclusions?

Reviewer #1: Yes

Reviewer #2: Yes

Reviewer #3: No

2. Has the statistical analysis been performed appropriately and rigorously? 

Reviewer #1: No

Reviewer #2: I Don't Know

Reviewer #3: I Don't Know

3. Have the authors made all data underlying the findings in their manuscript fully available?

Reviewer #1: Yes

Reviewer #2: Yes

Reviewer #3: Yes

4. Is the manuscript presented in an intelligible fashion and written in standard English?

Reviewer #1: No

Reviewer #2: No

Reviewer #3: No

5. Review Comments to the Author

Reviewer #1: Comments

Thank you for invitation to review Neonatal Hypothermia and Its Associated Risk Factors among Newborns at Arba Minch General Hospital, Ethiopia. The article is presented in a structured way and is well written. Below I provide some remarks on the manuscript parts. Additionally, I suggest the authors to discuss the implication of the result and include the ethical approval number.

1. In the abstract section, there is no convincing gap that shows why the research is needed?

2. In the abstract section, you should clearly indicate the design

3. I appreciate the authors use of multi-state Markov model to characterize the outcome variable.

4. In the introduction section, you need to show the previous works, how your finding adds to the existing knowledge, and gap of your work. Furthermore, you need to show the impact of the problem using current evidence. E.g., neonatal mortality due to hypothermia in Ethiopia

5. On the method part, it is too shallow and it didn’t address all important components of research methodology. Clearly indicate the design? You didn’t mention quality control measures. You need to show how you measure your outcome variable in detail.

6. On the result and discussion parts, I recommend the author to write pertinent findings only? I appreciate the authors trying different measurement of outcome variable. The discussion is well written and structured. However, it lacks the implication of the results. Try to use current evidences.

7. The paper needs language edition.

Reviewer #2: The manuscript appears to be well-written and comprehensive. It thoroughly covers all of the significant aspects of the scientific piece of work. Additionally, it satisfies the scientific and ethical requirements for publication. Moreover, the manuscript follows the Plos One reporting guidelines.

I should probably refrain from commenting on the statistical analysis conducted to identify characteristics linked to the shift of newborns from one state of hypothermia to the other because I'm not familiar with the model utilized.

In the manuscript, the authors have included all the data needed to support the main findings presented.

There are numerous grammatical, typographical, and word-ambiguity errors in the document, which requires editing. In the bracket before the issue under the specific section of the manuscript, I put my suggestion.

Variables in the Study

The response was considered hypothermia severity graded World Health Organization (WHO, 1997). In this study(add comma ,) the first two groups were combined and considered as (make it severe) sever or moderate

Estimated average time spent in transient state

According to Table 4, the estimated average length of time spent in severe or moderate hypothermia for newborns with very low birth weight was nearly 48 minutes, (add for) low birth weight was nearly 35 minutes, and normal birth weight was 24 minutes.

Result

The study included 391 newborns in total. Two hundred four (52.2 percent) of the newborns in this sample (make it was) were males, while 187 (47.8percent) were females.

The initiation of newborns to breastfeed within (make it two hours) two hour of birth was successful for 182 newborns (46.6percent), more than two hours of birth for 120 newborns (30.7percent) and 89 newborns (22.8percent) were not initiated to breastfeed at all and fed other milk.

Table 1 Physiological, Birth related, major admission diagnosis, and Behavioral factors associated with hypothermia among newborns at Arba Minch General Hospital, 2018-2020 (Row-2 Obstruct complication what does it mean)

Discussion

first paragraph: The difference could be due to the number of newborn babies with severe hypothermia (being) was small.

second paragraph: The infants who are most vulnerable to temperature change(make it changes), according to Elbaum et al., (is) are those (that are) who are smaller [21].

Review Comments to the Author

TOPIC:

Based on what is presented, the topic appears to be factors affecting transitioning from one type of hypothermia to another, however your title is "neonatal hypothermia and its associated risk factors." What do you think about this?

ABSTRACT:

Part of the abstract's method is lacking in terms of study design, sampling strategy, and statistical analysis.

Background

Don't you think it would be wise to discuss about the proposed model and its use in the background? you should also discuss factors associated to newborn hypothermia.

METHOD

What is the study design used? It is important to note the study design used for this study.

How do you calculate your sample size? You should clearly explain how you determined your sample size.

The method part lack clear definition and measurement outcome variable. Your outcome variable's definition and measurement must be specified in detail.

Variables in the study

The response was considered hypothermia severity graded World Health Organization (1997).In this study the first two groups were combined and considered as sever or moderate hypothermia (state 1), mild hypothermia (state 2), and normal body temperature (state 3).

First of all, the references should be updated; also, the WHOS classification for hypothermia severity does not appear to be followed by your classification system. Do you think this classification is clinically significant and useful?

RESULT

The initiation of newborns to breastfeed within two hours of birth was successful for 182 newborns (46.6percent).

Why do you set the early breastfeeding cutoff at two hours when it is actually less than one hour?

Table 1 Physiological, Birth related, major admission diagnosis, and Behavioral factors associated with hypothermia among newborns.

You should change the heading because the table shows a basic frequency distribution of newborn-related characteristics rather than any associations.

Tables should have headings, table numbers, years, and places to be self-explanatory; nevertheless, tables 2, 3, 4, and 6 are not correctly structured.

DISCUSSION

first paragraph

Relevant thermal deviations are likely to occur in a high proportion of newborns admitted with mild hypothermia, necessitating greater attention to thermal control of these subjects.

It is wise to start by presenting the key finding in the context of your objective because the first paragraph of the discussion appears as a conclusion.

It's also wise to substitute words like newborns for subjects.

Third paragraph

Our findings revealed that newborns with a very low birth weight, a low APGAR score, and a higher heart rate recovered from hypothermia more slowly. Neonatal hypothermia increases morbidity and length of stay in the hospital [20]. The infants who are most vulnerable to temperature change, according to Elbaum et al., are those who are smaller [21].

You mentioned two research, but you didn't say whether they supported or refuted your argument.

Fourth paragraph

The findings of the study revealed that the weight of the newborn baby had a significant effect on the rate of transition from severe or moderate hypothermia to mild hypothermia. The weight of the newborn baby, on the other hand, had a significant effect on the rate of transition from mild hypothermia to a normal state [9, 19].

This is a descriptive summary of other people's works; tell us about your findings, what makes them different or similar, and why.

Last paragraph

Our research found that newborn pulse rate was associated with the transition from mild hypothermia to severe or moderate hypothermia, newborn pulse rate was associated with transitions to the worse state. This study confirms previous findings that neonatal hypothermia is related to pulse rate [26]. A decrease in the infant's pulse rate is the most common symptom of hypothermia [27]. The assessment of pulse rates in newborn infants after birth is critical for

directing recovery efforts in the right direction

This comparison seems to be inapplicable since, in your study, pulse is linked to the change from mild to severe hypothermia, whereas it is linked to hypothermia in the latter study. How therefore should these two findings be compared?

CONCLUSION

It seems like your conclusion is an explanation of your result. It is a good idea to write a conclusion that takes into account the implications of the results and place them in a broader research context.

REFERENCES

Consider more contemporary literature as some of your references are out of date.

Some of your references have titles in italics, while others have journals in italics.Use a consistent layout across all your references.

Reviewer #3: Statistical analysis is based on a Markov Model and uncommon. The bases appear sound but I would recommend professional statistical advice.

English editing is necessary. Several misleading formulations (possibly due to English language deficiency) need correction

The discussion needs complete reformulation including limitations as most conclusions are not based on the findings.

6. PLOS authors have the option to publish the peer review history of their article (what does this mean?). If published, this will include your full peer review and any attached files.

Reviewer #1: No

Reviewer #2: No

Reviewer #3: **Yes: **Riccardo Pfister

---

## [Author Response · Author response to Decision Letter 0]

14 Nov 2022

First and foremost, we would like to express our gratitude for considering our paper, Risk Factors for Neonatal Hypothermia at Arba Minch General Hospital, Ethiopia by Tegenu Tento, Tilahun Asena, Meseret Alemayehu and Asmare Wube, for publication in your prestigious journal. We are now uploading the revised manuscript of the paper, which was prepared with all of the comments and suggestions of the respected referees in mind. We sincerely hope and believe that the revised manuscript will be acceptable for publication in your esteemed journal, BMC journal. The following are the Authors' point-by-point responses to the Referees' comments/suggestions for your consideration and action.

---

## [Decision Letter · Decision Letter 1]

25 Nov 2022

PONE-D-22-10601R1Risk Factors for Neonatal Hypothermia at Arba Minch General Hospital, EthiopiaPLOS ONE

Dear Dr. Asena,

Thank you for submitting your manuscript to PLOS ONE. After careful consideration, we feel that it has merit but does not fully meet PLOS ONE’s publication criteria as it currently stands. Therefore, we invite you to submit a revised version of the manuscript that addresses the points raised during the review process.In addition, the authors should correct the generated ethics statement (currently NA) to denote ethical clearance details (as mentioned in the manuscript).Please submit your revised manuscript by Jan 09 2023 11:59PM. If you will need more time than this to complete your revisions, please reply to this message or contact the journal office at plosone@plos.org. Please include the following items when submitting your revised manuscript:A rebuttal letter that responds to each point raised by the academic editor and reviewer(s). You should upload this letter as a separate file labeled 'Response to Reviewers'.A marked-up copy of your manuscript that highlights changes made to the original version. You should upload this as a separate file labeled 'Revised Manuscript with Track Changes'.An unmarked version of your revised paper without tracked changes. You should upload this as a separate file labeled 'Manuscript'.If applicable, we recommend that you deposit your laboratory protocols in protocols.io to enhance the reproducibility of your results. Protocols.io assigns your protocol its own identifier (DOI) so that it can be cited independently in the future. For instructions see: https://journals.plos.org/plosone/s/submission-guidelines#loc-laboratory-protocols. Additionally, PLOS ONE offers an option for publishing peer-reviewed Lab Protocol articles, which describe protocols hosted on protocols.io. Read more information on sharing protocols at https://plos.org/protocols?utm_medium=editorial-email&utm_source=authorletters&utm_campaign=protocols.

We look forward to receiving your revised manuscript.

Kind regards,

Elsayed Abdelkreem, MD, PhD

Academic Editor

PLOS ONE

Journal Requirements:

Reviewers' comments:

Reviewer's Responses to Questions

**Comments to the Author**

1. If the authors have adequately addressed your comments raised in a previous round of review and you feel that this manuscript is now acceptable for publication, you may indicate that here to bypass the “Comments to the Author” section, enter your conflict of interest statement in the “Confidential to Editor” section, and submit your "Accept" recommendation.

Reviewer #1: (No Response)

2. Is the manuscript technically sound, and do the data support the conclusions?

Reviewer #1: Yes

3. Has the statistical analysis been performed appropriately and rigorously? 

Reviewer #1: Yes

4. Have the authors made all data underlying the findings in their manuscript fully available?

Reviewer #1: Yes

5. Is the manuscript presented in an intelligible fashion and written in standard English?

Reviewer #1: No

6. Review Comments to the Author

Reviewer #1: Thank you for your effort in addressing the comments. The majority of the comments were addressed. However, the authors should critically consider revising the typo errors. Even you should be wise when writing words. Look at your response to authors cover letter “We sincerely hope and believe that the revised manuscript will be acceptable for publication in your esteemed journal, BMC journal”.

7. PLOS authors have the option to publish the peer review history of their article (what does this mean?). If published, this will include your full peer review and any attached files.

Reviewer #1: **Yes: **Addis Eyeberu

---

## [Author Response · Author response to Decision Letter 1]

5 Dec 2022

Response to Reviewers

Dear Editor,

We appreciate you and the reviewers for your precious time in reviewing our paper and providing valuable comments. It was your valuable and insightful comments that led to possible improvements in the current version. The authors have carefully considered the comments and tried our best to address every one of them. We hope the manuscript after careful revisions meet your high standards. The authors welcome further constructive Comments if any. Below we provide the point-by-point responses. All modifications in the manuscript have been highlighted.

Sincerely,

Tilahun Ferede, PhD

Feredetilahun14@gmail.com

Assistant Professor

Department of Statistics

Arba Minch University

Response to Reviewer

1. If the authors have adequately addressed your comments raised in a previous round of review and you feel that this manuscript is now acceptable for publication, you may indicate that here to bypass the “Comments to the Author” section, enter your conflict of interest statement in the “Confidential to Editor” section, and submit your "Accept" recommendation.

Reviewer #1: (No Response)

Response: Thanks for your comment. We revised the entire manuscript for comments to the author and revised all comments accordingly.

2. Is the manuscript technically sounds, and do the data support the conclusions?

Reviewer #1: Yes

Response: Thank you for your comments. We have gone through your comments carefully

and tried our best to address them one by one. We hope the manuscript has been improved accordingly.

3. Has the statistical analysis been performed appropriately and rigorously? 

Reviewer #1: Yes

Response: Thank you for your comments and responses.

4. Have the authors made all data underlying the findings in their manuscript fully available?

Reviewer #1: Yes

Response: Thank you very much for the comment. 

5. Is the manuscript presented in an intelligible fashion and written in Standard English?

Reviewer #1: No

Response: Thank you very much for the comment. We did our best to correct these mistakes

6. Review Comments to the Author

Reviewer #1: Thank you for your effort in addressing the comments. The majority of the comments were addressed. However, the authors should critically consider revising the typo errors. Even you should be wise when writing words. Look at your response to author’s cover letter “We sincerely hope and believe that the revised manuscript will be acceptable for publication in your esteemed journal, BMC journal”.

6. Review Comments to the Author

Reviewer #1: Thank you for your effort in addressing the comments. The majority of the comments were addressed. However, the authors should critically consider revising the typo errors. Even you should be wise when writing words. Look at your response to author’s cover letter “We sincerely hope and believe that the revised manuscript will be acceptable for publication in your esteemed journal, BMC journal”.

PLOS authors have the option to publish the peer review history of their article (what does this mean?). If published, this will include your full peer review and any attached files.

Do you want your identity to be public for this peer review? For information about this choice, including consent withdrawal, please see our Privacy Policy.

Reviewer #1: Yes: Addis Eyeberu

Response: Thank you very much for the comment.

---

## [Editor Report · Decision Letter 2]

6 Dec 2022

Risk Factors for Neonatal Hypothermia at Arba Minch General Hospital, Ethiopia

PONE-D-22-10601R2

Dear Dr. Asena,

We’re pleased to inform you that your manuscript has been judged scientifically suitable for publication and will be formally accepted for publication once it meets all outstanding technical requirements.

Kind regards,

Elsayed Abdelkreem, MD, PhD

Academic Editor

PLOS ONE
---

## [Editor Report · Acceptance letter]

12 Dec 2022

PONE-D-22-10601R2 

Risk Factors for Neonatal Hypothermia at Arba Minch General Hospital, Ethiopia 

Dear Dr. Asena:

I'm pleased to inform you that your manuscript has been deemed suitable for publication in PLOS ONE. Congratulations! Your manuscript is now with our production department. 

Kind regards, 

on behalf of

Dr. Elsayed Abdelkreem 

Academic Editor

PLOS ONE